# Role of Vitamin D in Liver Disease and Complications of Advanced Chronic Liver Disease

**DOI:** 10.3390/ijms23169016

**Published:** 2022-08-12

**Authors:** Federico Ravaioli, Alessandra Pivetti, Lorenza Di Marco, Christou Chrysanthi, Gabriella Frassanito, Martina Pambianco, Chiara Sicuro, Noemi Gualandi, Tomas Guasconi, Maddalena Pecchini, Antonio Colecchia

**Affiliations:** 1Gastroenterology Unit, Department of Medical Specialties, University Hospital of Modena, University of Modena & Reggio Emilia, 41121 Modena, Italy; 2Department of Medical and Surgical Sciences, University of Bologna, 40128 Bologna, Italy; 3Clinical and Experimental Medicine PhD Program, University of Modena & Reggio Emilia, 41121 Modena, Italy

**Keywords:** vitamin D, cirrhosis, advanced chronic liver disease, NAFLD, HCC, portal hypertension, nutrition

## Abstract

Vitamin D is a crucial nutrient with many pleiotropic effects on health and various chronic diseases. The purpose of this review is to provide a detailed report on the pathophysiological mechanisms underlying vitamin D deficiency in patients with chronic liver disease, addressing the different liver etiologies and the condition of advanced chronic liver disease (cirrhosis) with related complications. To date, patients with liver disease, regardless of underlying etiology, have been shown to have reduced levels of vitamin D. There is also evidence of the predictive role of vitamin D values in complications and progression of advanced disease. However, specific indications of vitamin D supplementation are not conclusive concerning what is already recommended in the general population. Future studies should make an effort to unify and validate the role of vitamin D supplementation in chronic liver disease.

## 1. Introduction

Vitamin D is a fat-soluble and secosteroid hormone with a pleiotropic effect on human health and many chronic diseases. This review aims to summarise the currently available evidence on the role of Vitamin D in the different liver aetiologies and hepatological clinical scenarios, from chronic liver disease to advanced liver disease (cirrhosis) and its complications; lastly, we would like to indicate when vitamin D supplementation would benefit these patients.

## 2. Data Sources and Searches

We searched English-language publications on MEDLINE, Ovid, In-Process, Cochrane Library, EMBASE, and PubMed until May 2022. Literature searches were performed using the following keywords: Vitamin D, Vit. D, liver diseases, chronic liver disease, advanced chronic liver disease, portal hypertension, nonalcoholic fatty liver disease, acute hepatitis, hepatitis, alcoholic liver disease, cholestasis, hepatocellular carcinoma, HCC.

## 3. The Biochemistry and Physiology of Vitamin

### 3.1. Sources of Vitamin D

The most important source of vitamin D is its synthesis in the skin as a product of sunlight exposure. The ultraviolet-B (UVB) radiation converts the 7-dehydrocholesterol (a cholesterol metabolite in the plasma membrane) into previtamin D3, which rapidly isomerises to vitamin D3 in the lower epidermis heat-dependent process [1]. Once synthesised, vitamin D3 is driven out of the plasma membrane into the extracellular space, where the vitamin D–binding protein is transported into the dermal capillary bed. Vitamin D synthesis is determined by the intensity of the ultraviolet irradiation (spectrum 280–320 UVB) and varies according to the season and latitude. At the same time, melanin in the skin blocks UVB, thus limiting D3 production, as do sunscreen and clothes. A small proportion of Vitamin D can also be acquired from the diet: oily fish, egg yolks, milk, shiitake mushrooms, cocoa and cocoa-based foods (in which the amount of vitamin D depends on the fungal contamination level during fermentation) [2], liver, or organ meats. The vitamin D from animal sources is cholecalciferol (D3), while ergocalciferol (vitamin D2) is produced in plants and fungi [3,4,5]. Except for fatty fish (such as sardines, herring, tuna, mackerel, salmon, and cod liver oil), the amount of vitamin D naturally present in food is negligible unless fortified, such as in milk [6].

### 3.2. Metabolism of Vitamin D

Vitamin D itself is not biologically active. It undergoes a 3-step activation process before interacting with its receptor: 25-hydroxylation, 1α-hydroxylation, and 24-hydroxylation. These three steps are catalysed by cytochromes P450 mixed-function oxidases (CYPs such as CYP2R1, CYP27A1, CYP27B1, and CYP24A1), enzymes that are placed in the endoplasmic reticulum and the mitochondria. The electron donor chain for the endoplasmic reticulum enzymes is the reduced nicotinamide adenine dinucleotide phosphate (NADPH)-dependent P450 reductase. The electron donor chain for the mitochondrial enzymes includes ferredoxin and ferredoxin reductase. Once synthesised, Vitamin D is transported in the blood by vitamin D binding protein (DBP) to the liver, where it is hydroxylated at C-25 to produce 25-hydroxyvitamin D3 [25(OH)D3]. 25(OH)D3, also known as calcidiol, is the primary circulating form of vitamin D (88% are bound to DBP) and has a long half-life of 2–3 weeks. Its serum concentration has been used as one of the most reliable biomarkers of vitamin D status [7]. The next step in the synthesis of vitamin D is 1α-hydroxylation (by CYP27B1 activity) with the production of 1,25-dihydroxy vitamin D (1,25(OH)2D), known as calcitriol, which is also bound to DBP (85%), with a half-life of just 4 h. Its synthesis principally occurs in the kidney’s proximal tubule and, to a lesser extent, in the epithelial cells in the skin, lungs, intestine, breast, and prostate, endocrine glands, macrophages, T and B lymphocytes, dendritic cells (DCs). Unlike the hepatic 25-hydroxylases, the renal 1α-hydroxylase is tightly controlled by three hormones: parathyroid hormone (PTH), 1,25(OH)2D itself, and fibroblast growth factor 23 (FGF23). PTH stimulates whereas 1,25(OH)2D and FGF23 inhibit CYP27B1. The last step of the metabolism is 24-hydroxylation: the 24-hydroxylase catabolises 1,25(OH)2D into calcitroic acid, a biologically inactive agent, excreted in the bile. Recently, an alternative pathway for vitamin D activation at least in keratinocytes has been identified, namely 20-hydroxylation of vitamin D by CYP11A1, which appears to have activity similar to 1,25(OH)2D, at least for some functions [8] (Figure 1).

### 3.3. Vitamin D Binding Protein (DBP)

Vitamin D binding protein is an α-macroglobulin of 58 kDa, almost exclusively synthesised in the liver, belonging to the albumin gene family, located on chromosome 4, remarkably homologous to albumin and α-fetoprotein. The primary function of DBP is to bind vitamin D metabolites and transport them from their site of synthesis to the receptor. Recently, the role of DBP has been investigated and seems to participate in the activation of macrophages and behave as an actin scavenger [9]. Furthermore, it is known that DBP is characterised by polymorphism (the most common are Gc1F, Gc1S, and Gc2); indeed, a marked variation in the Gc1F allele, which has the highest affinity for vitamin D, is frequent in pigmented and keratinised skin type populations [10].

### 3.4. Mechanism of Action of Vitamin D

The key player of this game is 1,25(OH)2D, which finally activates the vitamin D receptor (VDR). VDR then forms a heterodimer with the retinoid X receptor, acting as a transcription factor and binding to vitamin D response elements in the promoter region of target genes. VDR is widely expressed in the whole human body. When its ligand activates it, it directly or indirectly manages the expression of different genes influencing cell differentiation, proliferation, apoptosis, angiogenesis and immunomodulation (through the activation of T-lymphocytes). Vitamin D activity is not only linked to bone metabolism; it has a pleiotropic effect [7,11,12]. For instance, vitamin D has an anti-cancer activity: high-dose vitamin D and high levels of serum 25(OH)D reduce the risk of cancer and have an anti-inflammatory effect and inhibitory role against cancer cells (such as colorectal, pancreatic or prostate cancer) [13]. Another role of 1,25(OH)2D3 is to suppress HIF-1, both at protein and transcriptional levels, reducing the expression of VEGF in various human cancer cells (colon, prostate and breast cancers) [14]. Additionally, Vitamin D can repress the oncogenic activity of β-catenin by activating its receptor (VDR) and can induce cell differentiation [15]. Through the control growing factors synthesis, Vitamin D also helps the regulation of neurotrophin, differentiation and maturation of neural cells [16]. Finally, many immunologic cells express vitamin D receptors (VDR), suggesting that vitamin D has a regulating immune function [17].

### 3.5. Vitamin D Deficiency

Vitamin D deficiency is a serum 25(OH)D level <20 ng/mL. Levels of 30–50 ng/mL are considered optimal, while levels between 20 and 30 ng/mL are identified as vitamin D insufficiency, and levels <10 ng/mL are identified as severely deficient [18,19,20]. The prevalence ranges from 16% to 100%. It depends on ethnicity, gender, age, season and latitude [21,22]. Its deficiency is related to a major risk of death. Significantly, vitamin D deficiency occurs not only in elderly people and those with chronic medical issues but also in young and healthy people. Patients with liver diseases are frequently characterised by vitamin D deficiency. The liver is a central organ in the activation of vitamin D. Furthermore, vitamin D-binding protein (DBP), the most important carrier of vitamin D, is synthesised in the liver [22,23], so if the hepatic function is compromised (also through cytochrome reduction), the enzymes that activate vitamin D and DBP (such as CYP24A1) could be negatively affected. As a matter of fact, vitamin D deficiency is a current question throughout Europe. Strategies to prevent its deficiency aim to find the right compromise between sun exposure and skin cancer risk, with discussion of vitamin D supplements and/or food fortification [24] (Table 1).

## 4. Vitamin D in the Chronic Liver Diseases

### 4.1. Viral Hepatitis

A recent study [25] showed that in patients with chronic viral hepatitis, especially in chronic hepatitis B virus infection (CHB) patients, an upregulation of CYP24A1 is involved in vitamin D degradation. Thus, 25(OH)D insufficiency could result from an imbalance between the production and degradation of vitamin D. Moreover, low levels of 25-OHD may contribute to the pathogenesis of chronic liver diseases and their unfavourable outcomes, increasing inflammation and fibrosis [26,27]. The alteration of extracellular matrix composition is a turning point for developing hepatic fibrosis. Some authors have investigated a possible relationship between deficiency in vitamin D, increased matrix metalloproteinases 2 and 9 and alternated collagen IV degradation [28,29]. The binding of 25-OHD with its receptors on fibroblasts can avoid migration and reduce the inflammatory pathway of liver stellate cells [26].

As long as the pathophysiological mechanism of 25(OH)D insufficiency has not been fully discovered, its deficiency, instead of being a cause of liver disease progression, could be a biological marker, resulting in deteriorating liver health [30]. However, other studies point out that there were no improvements in the viral hepatitis outcome through vitamin D supplementation [31].

#### 4.1.1. Hepatitis B Virus (HBV)

Hepatitis B virus (HBV) infection represents a worldwide health problem, with more than 350 million people chronically suffering from HBV infection [32]. Several studies have demonstrated that 25(OH)D levels are lower in CHB patients than in healthy and that insufficient vitamin D reduces the suppression of HBV replication [33,34]. Calcitriol (1,25(OH)2D) directly inhibits HBV activity by targeting the HBV core promoter, interfering with viral protein production and HBV replication rate [35,36,37,38]. Patients with sufficient vitamin D levels also show a better virological response to anti-HBV therapy (nucleos(t)ide analogues) [22] due to the potential immunomodulatory ability of 25(OH)D inactivating host immune genes by its intracellular VDR pathways [31,39,40,41].

#### 4.1.2. Hepatitis Delta Virus (HDV)

Patients with chronic hepatitis D (CHD) have a higher risk of developing cirrhosis than those with chronic hepatitis B mono-infection because of the rapid progression of fibrosis [42]. Despite new drug trials, the unique, effective therapy is pegylated interferon (peg-IFN), even if sustained virologic response achieved approximately 30%. Only a few studies investigated the role of vitamin D in CHD. Still, it seems that deficiency of 25(OH)D may negatively affect morbidity, mortality, and disease progression [43].

#### 4.1.3. Hepatitis C Virus (HCV)

Hepatitis C virus (HCV) infection has been a major sanitary problem since the advent of direct-acting antivirals (DAAs), which have provided a >90% cure rate in HCV-positive patients, changing the natural course of the infection [44,45]. The curative treatment determines a reduced inflammation-fibrosis and a consequent improvement of hepatic function within a few months [46]. This amelioration of hepatic status might positively influence vitamin D synthesis, with a slow but progressive increase in serum vitamin D levels [47]. Before the DAAs-era, HCV therapy was based on IFN: various meta-analyses have demonstrated a better virological response in HCV-positive patients with high serum levels of vitamin D [48], but contradictory results have been found in the role of vitamin D supplementation on the virological response and the degree of liver fibrosis [45,49]. Murayama and colleagues have demonstrated that vitamin D supplementation, through its metabolite 25(OH)D3, improves the efficacy of IFN-based therapy by inhibiting HCV production [49] by reducing the expression of ApoA1 and ApoC3 mRNAs. In particular, apolipoproteins are involved in HCV infectivity titers and core antigen levels [50]. On the other hand, studies have shown no significant changes in serum markers of hepatic fibrogenesis after vitamin D restoration in patients who underwent DAA therapy [45].

The relationship between viral hepatitis and vitamin D has been extensively analysed. Vitamin D may positively affect the course of viral hepatitis therapy and the progression of liver diseases; nevertheless, the actual connections are not wholly understood [51]. Further randomised controlled trials are needed to confirm the efficacy of Vitamin D supplementation in antiviral treatment strategies in the actual DAAs-era.

### 4.2. Alcoholic Liver Disease (ALD)

Alcohol consumption is a significant risk factor for death and disability, and its use is associated with several acute and chronic diseases [52]. An alcohol weekly consumption above 14 units in women and 21 in men results in an increased risk of alcoholic liver disease (ALD). DSM-V introduced the term alcoholic use disorder (AUD), defined as a problematic pattern of alcohol use leading to clinically significant impairment or distress [53]. AUD can lead to ALD, whose spectrum varies from early or asymptomatic ALD to alcoholic steatohepatitis and cirrhosis [54].

In recent years, increasing interest has been shown in the role of micronutrients, including Vitamin D, in ALD pathogenesis and their possible therapeutic implications [55]. Several studies reported lower vitamin D serum levels in patients with ALD and other chronic liver diseases [20,56]. Compared to healthy patients, median vitamin D levels were significantly lower in ALD patients (9.6 ng/mL vs. 19 ng/mL) [57]. In a cohort of ALD patients (with alcohol consumption of 80 g or more per day for at least five years) collected by Anty et al., 96% of patients had an insufficient Vitamin D level (<30 ng/mL), 86.1% were deficient (<20 ng/mL), and 60.4% were severely deficient (<10 ng/mL) [58].

Vitamin D deficiency in ALD recognises multiple pathogenic factors: lower dietary uptake, altered hepatic hydroxylation of vitamin D, malabsorption, reduced production of DBP in the liver, reduced sun exposure, jaundice or chronic inflammation [59]. Furthermore, either in vivo or in vitro, it has been demonstrated that 1,25(OH)2D can reduce TNF-alfa production, leading to a downregulation of the harmful cascade in the liver after an incongruous intake of alcohol [57].

Although the role of these factors remains unclear in the pathogenesis of vitamin D deficiency in ALD, the vitamin D deficiency implications cannot be ignored from a prognostic point of view. A prospective study that involved 324 ALD patients showed a significant association between severe deficiency in 25(OH)D (<10 ng/mL) and higher aspartate aminotransferase (AST) levels, increased hepatic venous pressure gradient (HVPG), MELD and Child–Pugh (CP) scores. Furthermore, in this cohort, patients severely deficient in vitamin D showed increased mortality at one year [57].

The therapeutic implications of vitamin D supplementation in ALD are not fully known. Malham et al. demonstrate that administering a single oral dose of ergocalciferol or cholecalciferol could temporarily raise plasma levels of 25(OH)D to normal values in approximately 50% of the patients with alcoholic cirrhosis recruited in the study. They also demonstrate that the biological availability of 25(OH)D is higher in patients with mild liver dysfunction (CP-A and B) compared with patients with severe liver disease (CP-C), showing that the degree of liver dysfunction affects the ability to increase 25(OH)D levels after treatment [59]. Conversely, Savic et al. analysed a cohort of 70 patients with alcoholic cirrhosis treated at a cholecalciferol daily dose of 1000 UI. Patients with CP-C experienced the highest increase in vitamin D levels during the study’s follow-up period in this cohort. They also showed a decrease in CP score at one year. However, it must be considered that alcohol withdrawal may ameliorate liver function and it is not only linked to vitamin D supplementation [60].

ALD patients showing a vitamin D deficiency may also manifest alterations in bone metabolism resulting in osteoporosis, osteomalacia, increased risk of fractures and delayed fracture healing. Factors contributing to bone disease in alcoholic patients include the direct effect of ethanol, increased risk of trauma, malnutrition, alcoholic neuropathy, and increased proinflammatory cytokines [61]. The treatment targets increase bone formation and reduce bone degradation through vitamin D and calcium supplementation [62]. Moreover, abstinence from alcohol intake is mandatory for preventing bone loss, and physical activity is also recommended [62,63].

Vitamin D is deficient in patients with ALD due to several mechanisms. Low Vitamin D serum levels lead to alterations in bone metabolism and seem to correlate to a poor prognosis of ALD-related chronic liver disease. Further studies on larger cohorts are required to define vitamin D’s pathogenetic and prognostic role in ALD and its supplementation role in all-cause mortality, liver-related mortality and overall survival.

### 4.3. Non-Alcoholic Fatty Liver Disease (NAFLD)

Non-alcoholic fatty liver disease (NAFLD) is the most common chronic liver disease worldwide, and health and economic systems will be affected by its evolution in the future [64]. According to histological analysis, NAFLD is characterised by excessive hepatic fat accumulation, associated with insulin resistance and defined by the presence of steatosis in >5% of hepatocytes [65]. The diagnosis of NAFLD requires excluding secondary causes and high alcohol consumption [66].

For the linkage between metabolic diseases and NAFLD, it was recently proposed to replace NAFLD with Metabolic dysfunction Associated with Fatty Liver ± Disease (MAFL/MAFLD) [67], as well as to set up new criteria for a positive diagnosis of MAFLD involving overweight/obesity, presence of type 2 diabetes mellitus (T2DM), or evidence of metabolic dysregulation, in addition to histological (biopsy), imaging or blood biomarker evidence of fat accumulation in the liver [68].

In recent decades, experimental evidence has proven the involvement of vitamin D in many immune-inflammatory and metabolic processes, particularly the active form of 1,25(OH)2D and the VDR [69,70]. Vitamin D properties have been experimentally demonstrated at the hepatic level, with direct insulin-sensitizing [71], anti-inflammatory and anti-fibrotic actions [72]. Low vitamin D levels have been associated with insulin resistance-related diseases, such as diabetes, metabolic syndrome, and MAFLD [73]. Two recent cross-sectional studies showed that Vitamin D deficiency is associated with a higher risk of steatosis, represented by the Fatty Liver Index (FLI) score in obese patients [74]; in women, this association persists regardless of metabolic profile and body weight [75]. A retrospective cohort study shows that higher serum vitamin D levels were associated with a decreased risk of controlled attenuation parameter (CAP)-defined NAFLD, compared to low levels of serum vitamin D [76].

Increased free fatty acid and adipose tissue and a decrease of adiponectin lead to the development of insulin resistance in NAFLD [77]. The fundamental mechanism between vitamin D and insulin resistance is still unknown. The major vitamin D storage site in the body is adipose tissue, which produces adipokine and cytokines, which are involved in generating systemic inflammation [78]. In addition, current evidence suggests that vitamin D regulates insulin secretion of pancreatic β-cells [79].

Given such evidence, randomized controlled trials (RCTs) have found that additional vitamin D treatment may improve insulin resistance, marked by a decrease in Homeostasis Model Assessment Insulin Resistance (HOMA-IR) in patients with NAFLD [80,81,82]. These studies were included in a recent meta-analysis that analysed insulin resistance and serum ALT levels [83]. In a double-blinded, randomised, placebo-controlled trial, vitamin D3 treatment (50,000 IU for 12 weeks) improved HOMA-IR, serum ALT, AST, PCR, and adiponectin but without effect on body weight or serum lipids [84]. Moreover, ten independent trials were included in a meta-analysis that provided substantial evidence that vitamin D could be an adjunct in the pharmacotherapy of NAFLD [85]. On the other hand, in another prospective study of 13 patients with NASH who underwent supplementation with high doses of vitamin D3 (25.000 IU for 24 weeks), there were no changes in liver enzymes, HOMA-IR, adipocytokine profiles or liver histology [86]. A recent meta-analysis found that vitamin D supplementation did not affect FPG, insulin, HOMA-IR, triglycerides, total-, LDL- and HDL-cholesterol, AST, ALT levels, and BMI [87].

In conclusion, clinical trials do not report unequivocal beneficial effects of vitamin D supplementation on liver markers in patients diagnosed with NAFL/MAFLD. Moreover, the investigations so far involved small populations and were heterogeneous regarding inclusion criteria and outcome setting. Nonetheless, the evidence shows positive effects of long-term low-dose vitamin D treatment in the youngest populations of MAFLD subjects, without advanced fibrosis and comorbidities, such as diabetes [70]. We need further studies on larger populations with standardised criteria to define the potential role of vitamin D supplementation in patients with fatty liver disease.

### 4.4. Autoimmune and Cholestatic Liver Disease

#### 4.4.1. Vitamin D and Autoimmunity

It is now well established that Vitamin D’s immunomodulatory and anti-inflammatory role is based mainly on its effect on immune cells through the VDR [88]. VDR also plays a role in immune cell differentiation and proliferation [89], and some VDR polymorphisms have been associated with primary biliary cholangitis and autoimmune hepatitis. [89] Since autoimmune disease incidence appears to be higher in northern countries, where sunlight exposure is lower, there seems to be a connection between low levels of Vitamin D and autoimmune diseases [88,90]. Low levels of Vitamin D have been correlated with numerous autoimmune diseases [88], such as type 1 diabetes mellitus, systemic lupus erythematosus [2], rheumatoid arthritis and, most notably, multiple sclerosis [3]. By contrast, little is known about the association between autoimmune liver diseases (autoimmune hepatitis, primary biliary cirrhosis and primary sclerosis cholangitis) and vitamin D.

#### 4.4.2. Autoimmune Hepatitis

Autoimmune hepatitis (AIH) is an immune-mediated liver disease mainly affecting young Caucasian women. AIH can be asymptomatic, subclinical or responsible for acute liver failure. Therapy is based on glucocorticoids alone or in combination with azathioprine [91]. In particular, according to recent European Association for the Study of the Liver [91] guidelines, first-line treatment is based on prednisolone (initial dosage of between 0.5 and 1 mg/kg/day) followed, after two weeks, by the addition of azathioprine (initial dosage of 50 mg/day, gradually increased to a maintenance dose of 1–2 mg/kg). This dual therapeutic approach, even though it has not demonstrated superiority compared to steroid therapy alone in inducing remission of autoimmune hepatitis, lowers the incidence of steroid side effects [91].

A few studies have explored the relationship between AIH and vitamin D. Efe et al. [92] found that mean vitamin D levels were significantly lower in 68 AIH patients than in controls (16.8 ± 9.2 vs. 35.7 ± 13.6, *p* < 0.0001) [4]. This finding was confirmed in a more recent study by Abe et al. [93], in which they also highlighted that serum total vitamin D levels were significantly lower in patients with acute AIH than in those with chronic AIH (13.2 ng/mL vs. 16.0 ng/mL, *p* = 0.029). [3] A study showed that serum total vitamin D levels were significantly lower in patients with severe necro-inflammatory activity (12.4 ng/mL vs. 16.8 ng/mL; *p* = 0.0017) [93]; by contrast, levels of pro-inflammatory cytokines (IFN-γ and IL-33) were significantly higher in AIH patients with serum total Vitamin D levels of <15 ng/mL (IFN-γ: 0.21 vs. 0 pg/mL, *p* = 0.0181; IL-33: 26.54 vs. 13.13 pg/mL, *p* = 0.0036).

Vitamin D level was also correlated to histological AIH features. Efe et al. [92] demonstrated that fibrosis scores and severe interface hepatitis were independently and negatively associated with low Vitamin D levels (*p* = 0.014; OR 0.12, 95 % CI 0.02–0.65 and *p* = 0.020, OR 0.17, 95 % CI 0.04–0.76, respectively). In light of these findings, they suggested vitamin D as a marker to predict histological features of AIH and severe liver disease AIH-related [92]. Namely, Ebadi et al. [94] illustrated how patients with AIH and severe vitamin D deficiency appeared to have higher MELD scores, lower serum albumin levels, greater frequency of cirrhosis at presentation, risk of death for liver-related causes, and need for liver transplantation in comparison to patients with higher levels of Vitamin D.

Finally, three studies [92,93,94], found a correlation between low vitamin D levels and less probably response to therapy, with a higher number of patients needing a combination therapy (steroids + azathioprine) when Vitamin D levels were <15 ng/mL. These findings may correlate to vitamin D’s ability to promote the anti-inflammatory action of glucocorticoids [95].

#### 4.4.3. Primary Biliary Cholangitis (PBC) and Primary Sclerosing Cholangitis (PSC)

Ebadi et al. [96] evaluated that, as shown for AIH, vitamin D deficiency is significantly more common in patients with PBC than in healthy controls. These results were confirmed by Agmon-Levine et al. [97].

PBC is a Th1-mediated-autoimmune disease [96] e, and VDR agonists appear to inhibit proinflammatory, pathogenic T cells such as Th1 and Th17 cells and favour the development of Th2 and T regulatory cells [98]. Therefore, it seems reasonable to investigate a link between VDR and PBC susceptibility. Multiple studies demonstrated a correlation between VDR polymorphisms and susceptibility to primary biliary cholangitis in different ethnic groups [99,100,101,102]. In particular, Tanaka et al. [99] found a significant association in the frequency of VDR polymorphism (BsmI) between Japanese and Italian patients with PBC and controls. A Hungarian study also confirmed this result [100]. On the other hand, German [102] and Chinese [101] studies disclosed a protective VDR polymorphism (BsmI) association. The association between PBC susceptibly and VDR polymorphisms remains controversial, as highlighted by a recent metanalysis by Mo et al. [103].

Moreover, low levels of vitamin D appeared to be associated with incomplete response to UDCA therapy andto be a marker of disease severity. Namely, based on biopsy results and bilirubin levels, Agmon-Levine et al. [97] found a significant correlation between low vitamin D levels and advanced liver damage in PBC patients. Finally, liver-related outcomes, such as mortality and liver transplantation, were more frequent in patients with low vitamin D levels. Liver transplantation indication in PBC was more common in patients with vitamin D deficiency (20% vs. 6%, *p* = 0.003) and the 5- and the 10-year probability of liver-related event-free survival was significantly higher in patients without vitamin D deficiency than in those with abnormal values (82% and 73% vs. 97% and 95%, log-rank, *p* < 0.001) [96].

Little is known about the correlation between serum vitamin D levels and primary sclerosing cholangitis (PSC). Jorgensen et al. evaluated fat-soluble vitamin levels in 56 patients with PSC and 87 patients with advanced PSC studied for liver transplantation and found a direct correlation between the severity of the PSC and serum vitamin D levels [104].

In conclusion, current evidence suggests a tight correlation between autoimmune and cholestatic liver diseases and low levels of Vitamin D. However, further studies are needed to establish either the causative role of Vitamin D deficit in autoimmune liver disease, thus identifying a potential new therapeutic target, or its applicability as an innovative marker of both severe and advanced autoimmune liver disease.

## 5. Vitamin D in Advanced Chronic Liver Disease (ACLD) and Complications

Several studies have shown that vitamin D deficiency is a common feature in those suffering from ACLD, independently of the aetiology [105]. It has been estimated that the incidence of vitamin D insufficiency in cirrhotic patients is between 64 and 92%, higher than in the general population [20]. Different factors can explain vitamin D deficiency in ACLD. Patients affected by ACLD have reduced exposure to vitamin D sources through low sunlight exposure and malnutrition; moreover, there is a low intestinal hormone absorption and a decreased production of binding proteins (DBP and albumin), which are responsible for transferring vitamin D to the liver and kidney so it can be activated. Lastly, a non-effective hydroxylation of vitamin D leads to lower levels of active hormones, while the catabolism of the vitamin is increased [106].

The correlation between the severity of ACLD and the deficiency of vitamin D could be explained by the fact that vitamin D improves portal hypertension through the activation of Vitamin D receptor (VDR) and Calcium-sensing receptor (CaSR) in a cirrhotic rats model [107]. The severity of the ACLD by CP score and HVPG values is linked with decreasing levels of Vitamin D; Paternostro et al. proposed vitamin D deficiency as a valuable predictor of mortality in cirrhosis [108]. Also, Putz-Bankuti et al. showed that vitamin D deficiency is highly prevalent in a cohort of 75 consecutive ACLD patients and that low 25(OH)D levels are associated with liver dysfunction assessed by CP and MELD scores. A vitamin D deficit was also a predictor of overall mortality and hepatic decompensation in ACLD patients [109]. Based on all this evidence, however, a recent Cochrane meta-analysis by Bjiealovich et al. concluded that there is a lack of evidence to demonstrate the efficacy of vitamin D supplementation in reducing the causes of mortality and morbidity of ACLD and that the overall quality of evidence is very low [110].

### 5.1. Hepatic Encephalopathy

Kumar et al. analysed the correlation between vitamin D deficiency and hepatic encephalopathy (HE) [111]. They conducted a cross-sectional study in a cohort of 100 subjects of the North Indian population, which demonstrated that the mean level of 25(OH)D was significantly lower in subjects with HE versus the control group (25.62 ± 21.94 nmol/L vs. 37.44 ± 18.61 nmol/L, *p* < 0.001) and significantly stratified for HE grades (*p* < 0.0001): 30.64 ± 21.64 nmol/L in grade 1 HE, 12.03 ± 11.05 nmol/L in grade 3, and 18.8 ± 16.88 nmol/L in grade 4. Vidot et al. confirmed a statistically significant correlation between low 25-OHD levels and overt HE (OHE). Further studies are necessary to evaluate the role of vitamin D deficit in populations with covert HE [112]. Moreover, vitamin D level is suggested as an index of survival in the study of Afifi et al. They demonstrate that vitamin D > 18.5 nmol/L can predict survival in HE patients with a sensitivity of 94% and a specificity of 80% [113].

### 5.2. Gastro-Esophageal Varices and Variceal Bleeding

A study over 83 patients hospitalised from November 2016 to January 2017 demonstrated how serum 25(OH)D3 level was significantly lower in Esophageal Variceal Bleeding (EVB) cirrhotic patients and an independent risk factor for EVB [114]. Further studies are needed to clarify the role of vitamin D deficiency in esophageal bleeding. The presence of esophageal varices is considered a contraindication for using bisphosphonate therapy to treat osteoporosis. Lima et al. demonstrated that risedronate might be safely used in patients with varices at low risk of bleeding with a similar incidence of adverse digestive effects in the treatment and control group [115].

### 5.3. Bacterial Infections and Spontaneous Bacterial Peritonitis

Studies have also demonstrated the correlation between bacterial infections and vitamin D levels. Anty et al. assessed the vitamin D levels in cirrhotic patients with and without bacterial infection identifying a higher incidence of infections in patients with a severe vitamin D deficiency than in others (54 vs. 29%), and severe vitamin D deficiency was an independent predictor of infection [116].

Spontaneous bacterial peritonitis (SBP) is one of cirrhotic patients’ most frequently diagnosed infections. Vitamin D deficiency is prevalent in SBP cirrhotic patients and is an independent predictor of infection and death. The serum ascites vitamin D gradient (SADG) has been proposed as an index in SBP [117]. Hafez et al. investigated 88 patients with ascites determining that SADG ≥ 5.57 ng/mL had a sensitivity of 70.5% and a specificity of 68.2% for exclusion of SBP [118].

The VDR system and its downstream gene LL-37 are involved in the pathogenesis and antibacterial immune response to SBP. Zhang et al. showed how the expression of peritoneal leucocytes VDR and LL-37 genes were simultaneously up-regulated in cirrhotic patients with SBP compared with those with uncomplicated ascites [119].

Moreover, vitamin D levels are correlated with the effectiveness of the therapy of SBP and with the survival rate after SBP; in particular, Vitamin D supplementation was associated with improvement in their response to treatment. Indeed, Mohamed et al. randomly stratified groups of patients with SBP and vitamin D deficiency into treatment and control groups. Over six months of treatments, they found a significant improvement in serum vitamin D level in the group supplemented with vitamin D along with a statistically significant increase in survival rate (64% vs. 42%; *p* < 0.05) and life duration (199.5 days vs. 185.5 days; *p* < 0.05). Univariate and multivariate regression analysis confirmed that Vitamin D supplementation was positively correlated with survival over six months (*p* < 0.001; adjusted hazard ratio: 0.895) in patients with ascites and SBP [120].

### 5.4. Sarcopenia

Sarcopenia is a common complication in chronic diseases, and consists of losing skeletal muscle volume [121,122]. In ACLD, there is a direct correlation between the grade of sarcopenia and the severity of liver dysfunction [123,124].

Okubo et al. conducted a prospective randomised controlled pilot trial to demonstrate the role of vitamin D in restoring the skeletal muscle volume. They found that lower serum 25(OH)D levels were significantly and independently related to sarcopenia in patients with ACLD. Moreover, the authors found that the cirrhotic patients consuming branched-chain amino acids and vitamin D achieved a more significant increase in the skeletal mass index (+5.8% per year) compared with the group that consumed only branched-chain amino acids [125]. Further research is needed to better understand the effect of vitamin D supplementation on sarcopenia treatment, even more so in ACLD patients [126].

All the studies we considered show a correlation between ACLD complications and vitamin D levels. Several data sources are available on HE, infection, and sarcopenia, while information on the association between vitamin D deficiency and risk of variceal bleeding is lacking. However, the role of vitamin D in the pathogenesis of ACLD complications remains unclear and needs further investigation.

## 6. Vitamin D and HepatoCellular Carcinoma (HCC)

Vitamin D deficiency occurs in 90% of patients with hepatocellular carcinoma (HCC) and constitutes an epiphenomenon in the setting of concomitant liver disease in an advanced stage (ACLD) [127]. In recent years, the anticancer properties of vitamin D, also in the context of HCC, have been demonstrated [128]. Preclinical evidence suggests that vitamin D deficiency may facilitate the progression of HCC and other types of tumours. The meta-analysis by Gaksch et al. proposed an inverse relationship between the serum value of vitamin D and the risk of developing HCC [129,130]. The increased bioavailability of circulating 25(OH)D3 is also associated with the survival of patients with HCC compared to total or free vitamin levels [131]. In contrast, Liu et al. [132] report that higher values of 25(OH)D are associated with an increased risk of HCC; however, genetic variations related to the metabolism of vitamin D can influence tumour response and survival [132].

### 6.1. Vitamin D in Hepatocarcinogenesis

The role of vitamin D in proliferation, angiogenesis, apoptosis, inflammation and cell differentiation is known, and the anti-inflammatory role of vitamin D in carcinogenesis has already been demonstrated [133]. Generally, inflammation induces oxidative stress by activating neutrophils and Kupffer cells, stimulating carcinogenesis [134]. Oxidative stress is inexorably associated with the onset and progression of HCC [134]. Studies report that cholecalciferol (25(OH)D3) could be involved in controlling oxidative stress and, therefore, protecting against carcinogenesis [135]. Furthermore, 25(OH)D3 appears to be able to downregulate the expression of tumour growth factor β (TGFβ) by activating caspase 3 [136] and restoring its expression, initially lost in the liver tumours [137]. 1,25(OH)2D also inhibits the formation of new blood vessels, preventing the neoangiogenesis underlying hepatocarcinogenesis mediated by the vascular endothelial growth factor (VEGF) [138]. Several studies reported the inverse relationship between serum levels of 1,25(OH)2D and the risk of developing HCC [128]. They suggested how the overall survival (OS) in patients with HCC is significantly lower in patients with concentrations of 1,25(OH)2D <10 ng/mL [139]. Furthermore, 1,25(OH)2D deficiency appears to be associated with more advanced stages of HCC and, consequently, a worse prognosis [140]. Ultimately, some in vivo studies showed that vitamin D could regulate the progression of HCC through the activation of apoptosis, reducing oxidative stress and inflammation [137]. Many studies have reported the relationship between VDR polymorphisms and the onset of HCC [141,142]; for example, a Chinese study has shown that the presence of the VDR rs2228570 and DBP rs7041 polymorphisms are linked to an increased incidence of HCC in patients with HBV infection [143]. Similarly, Falleti et al. showed that VDR polymorphisms are associated with the onset of HCC in patients with alcoholic cirrhosis [141]. It has also been reported that HCC patients with HCV-related liver disease have lower vitamin D values than healthy individuals [142].

Clinical trial data is still lacking despite in vitro and in vivo studies (Table 2).

### 6.2. Vitamin D as Biomarkers in HCC

Several studies have shown the role of the alteration of VDR methylation in various neoplasms [144,145,146]; the study of Mai Abdalla [147] evaluated the methylation of the VDR promoter as a potential biomarker for the identification of HCC. Notably, it was shown that VDR promoter methylation expression in patients with HCC was positively associated with AFP values [132]; in addition, the combined use of VDR gene promoter methylation with the AFP routinely increased its sensitivity to 93.3% [132].

### 6.3. Vitamin D Supplementation Therapy in HCC

The potential therapeutic role of vitamin D supplementation is limited by the risk of dangerous hypercalcemia; therefore, in recent years, vitamin D analogues have been studied to reduce this risk. Indeed, Dalhoff et al. investigated seocalcitol in an uncontrolled phase II study which showed anti-tumorigenic properties with transitory hypercalcemia [148]. In a small phase, an analysis of six patients, Finlay et al. demonstrated how administering 1,25(OH)2D by continuous hepatic arterial infusion allows administering a high dosage without inducing hypercalcemia [149].

In a recently published study [150] in 58 patients with hepatocellular carcinoma, candidates for transarterial chemoembolisation (TACE), 25(OH)D was measured at baseline and the day after TACE and the lowest vitamin D levels at baseline correlated with the worst post-TACE result [150]. However, it should be considered that at baseline almost half of the enrolled patients had a vitamin D deficiency, probably due to the ACLD status [150]. Data from Morris et al. [151] support the hypothesis that, in patients with HCC, the locoregional administration of 1,25(OH)2D in lipiodol may allow supra-pharmacological doses of the drug without the development of hypercalcemia [18]. Matsuda’s study [152]; evaluated the use of vitamin D and its analogues in HCC β-catenin mutated synthetic transgenic mice [152]. Although 1,25(OH)2D is reported to inhibit the β-catenin signal, as well as tumour cell proliferation, migration, and differentiation, neither the use of vitamin D nor its analogues in this study showed an advantage in terms of OS or progression of HCC [152].

In conclusion, actual data concerning the association between vitamin D, VDR, and HCC, in addition to attempting to explain the mechanisms related to the role of vitamin D in HCC, from a pathogenetic and therapeutic point of view, could open new horizons on the role of hormones in hepatocarcinogenesis, and also to associate vitamin D analogues with standard therapy for HCC.

## 7. Conclusions and Future Perspective

The pleiotropic effects of vitamin D have been demonstrated in several chronic and acute liver diseases. As described in our review, much effort has been made in recent years to investigate for which liver conditions vitamin D deficiency is clinically relevant and when its supplementation would lead to clinical benefit for patients.

In brief, Figure 2 summarises the main findings on the role of vitamin D in various chronic liver diseases and liver complications. 

Nevertheless, the overall quality of such evidence is low, and even when meta-analysis studies are available, the results have not always led to a firm conclusion.

In particular, vitamin D deficiency is accentuated in patients with CLD, and also found in the general population, especially in the winter seasons. Therefore, restoration to the normative range of values should be applied in these patients. Until there is sufficient evidence from prospective studies and hopefully future RCTs, it may be worth using overall clinical judgment by anticipating vitamin deficiency, as well as that of other micronutrients, with laboratory screening and then supplementation only in case of proven deficiency.

We believe that until there is sufficient evidence from prospective studies and hopefully future RCTs, it may be worth applying general clinical judgment by anticipating vitamin deficiency, as well as other micronutrients, with laboratory screening and supplementation only in the case of a proven deficit.

## Figures and Tables

**Figure 1 ijms-23-09016-f001:**
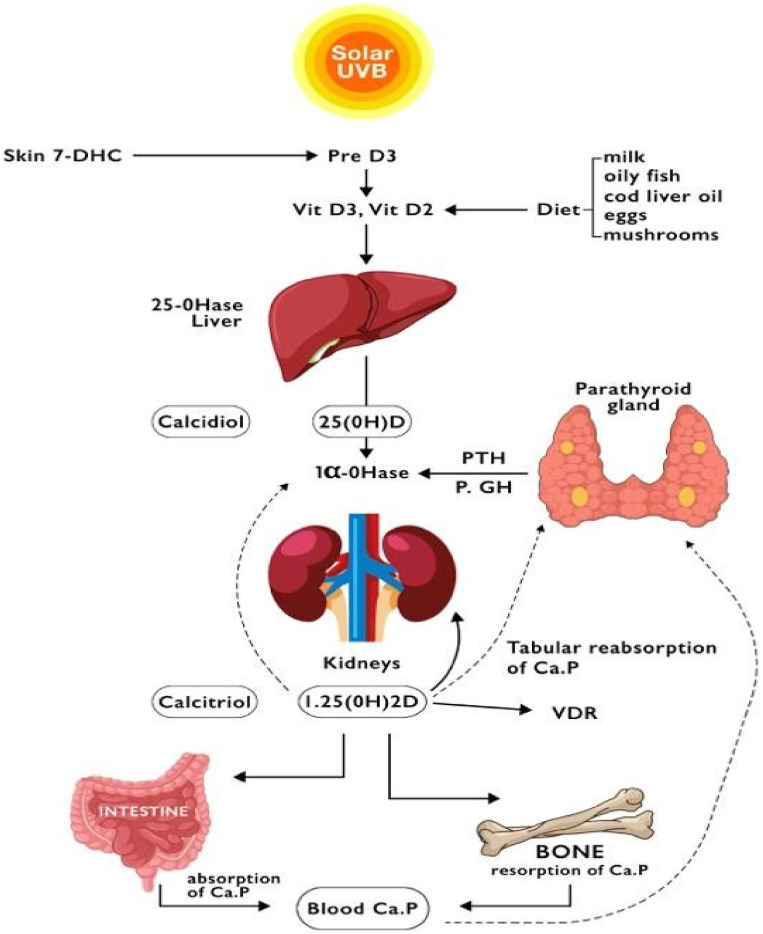
The metabolism of Vitamin D. Solid arrows show the direct effects of its products, and dotted lines demonstrate the negative feedback of plasma calcium or 1,25(OH)2D. (Ca: calcium; 7-DHC: 7-dehydrocholesterol; GH: growth hormone; 1a-OHase: 1-alpha-hydroxylase; 25-OHase: 25-hydroxylase; P: phosphate; PTH: parathyroid hormone; VDR: vitamin D receptor; Vit: vitamin).

**Figure 2 ijms-23-09016-f002:**
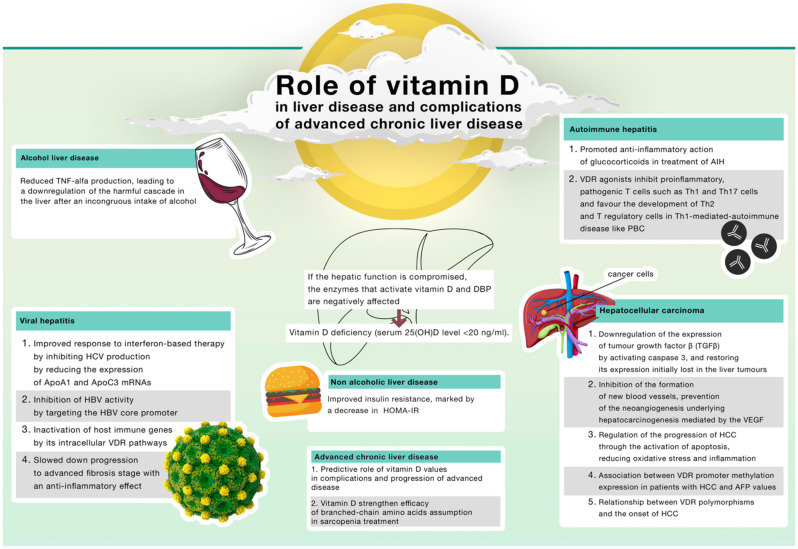
Role of Vitamin D in chronic liver diseases. (TNF: tumour necrosis factor; AIH: autoimmune hepatitis; VDR: vitamin D receptor; Th: T helper; PBC: primary biliary cholangitis; HCV: hepatitis C virus; ApoA1: Apolipoprotein A1; ApoC3: apolipoprotein C-III; HBV: hepatitis B virus; HOMA-IR: Homeostasis Model Assessment Insulin Resistance; VEGF: vascular endothelial growth factor; HCC: hepatocellular carcinoma; AFP: alpha-fetoprotein).

**Table 1 ijms-23-09016-t001:** Impact of vitamin D supplementation in liver and non-liver diseases^1^ (HOMA-IR: Homeostasis Model Assessment Insulin Resistance; HCV: hepatitis C virus; ApoA1: Apolipoprotein A1; ApoC3: apolipoprotein C-III; HBV: hepatitis B virus; VDR: vitamin D receptor; NAFLD: non-alcoholic fatty liver disease; ALD: alcoholic liver disease; TNF: tumour necrosis factor; AIH: autoimmune hepatitis; Th: T helper; PBC: primary biliary cholangitis; SBP: spontaneous bacterial peritonitis; ACLD: advanced chronic liver disease; HCC: hepatocellular carcinoma; TGF: transforming growth factor; VEGF: vascular endothelial growth factor; AFP: alpha-fetoprotein).

Target	Action
**Non-Liver Diseases**	
IMMUNE SYSTEM	Reduced risk of respiratory infection, such as tuberculosis and COVID-19 and sepsis.
	Improved response to steroid treatment in autoimmune disease, like psoriasis, type 1 diabetes, multiple sclerosis, rheumatoid arthritis.
INSULIN SENSITIVITY	Better control of insulin secretion of pancreatic β-cell. Improved insulin resistance, marked by a decrease in HOMA-IR
CARCINOGENESIS	Reduced risk of breast, colon, pancreatic and prostate cancer
**Liver Disease**	
VIRAL HEPATITIS	Improved response to interferon-based therapy by inhibiting HCV production by reducing the expression of ApoA1 and ApoC3 mRNAs
	Inhibition of HBV activity by targeting the HBV core promoter
	Inactivation of host immune genes by its intracellular VDR pathways
	Slowed down progression to advanced fibrosis stage with an anti-inflammatory effect
NAFLD	Improved insulin resistance, marked by a decrease in Homeostasis Model Assessment Insulin Resistance (HOMA-IR)
ALD	Reduced TNF-alfa production, leading to a downregulation of the harmful cascade in the liver after an incongruous intake of alcohol
AIH	Promoted anti-inflammatory action of glucocorticoids in treatment of AIH
	VDR agonists inhibit proinflammatory, pathogenic T cells such as Th1 and Th17 cells and favour the development of Th2 and T regulatory cells in Th1-mediated-autoimmune disease like PBC
ACLD	Antibacterial immune response to SBP of VDR system and its downstream gene LL-37
	More significant increase in the skeletal mass index compared with only branched-chain amino acids assumption in sarcopenia treatment
HCC	Downregulation of the expression of tumour growth factor β (TGFβ) by activating caspase 3, and restoring its expression initially lost in the liver tumours
	Inhibition of the formation of new blood vessels, prevention of the neoangiogenesis underlying hepatocarcinogenesis mediated by the VEGF
	Regulation of the progression of HCC through the activation of apoptosis, reducing oxidative stress and inflammation
	Relationship between VDR polymorphisms and the onset of HCC
	Association between VDR promoter methylation expression in patients with HCC and AFP values

**Table 2 ijms-23-09016-t002:** Summary of clinical trials from ClinicalTrials.gov.

Identifier	Study Title	Study Tipe	Intervention	Status
NCT01575717	The Effect of Vitamin D Repletion in Patients with Hepatocellular Carcinoma on the Orthotopic Liver Transplant List Interventional	Interventional	Drug: Vitamin D3 4000 IU Drug: Vitamin D3 2000 IU	Unknown
NCT02779465	Oral Vitamin D Treatment for the Prevention of Hepatocellular Carcinoma Interventional	Interventional	Drug: Vitamin D3	Not yet recruiting
NCT01956864	Study of High-Dose Oral Vitamin D for the Prevention of Liver Cancer Interventional	Interventional	Drug: Vitamin D	Withdrawn
NCT02461979	The Role of the Vitamin D Receptor Gene Polymorphisms in Hepatocarcinogenesis Interventional	Interventional	Other: The VDR genotype	Recruiting
IU: International Unit				

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
