# Peer review of "Role of Vitamin D in Liver Disease and Complications of Advanced Chronic Liver Disease"

_ijms, 2022, doi:10.3390/ijms23169016_

Round 1

Reviewer 1 Report

The manuscript is well organized and easy to read, covers many aspects, and even the figures and tables are well organized.

As pointed out by the authors, there are no studies that clarify whether vitamin D supplementation is necessary and/or beneficial for various diseases.

Probably, it should be emphasized that very often in the general population, especially in the winter seasons, a deficiency of vitamin D is noted; therefore, being some actions being well established (immune system in primis, but also inflammatory state), the correlated positive actions on the pathologies may be due only to the restoration of "normal" values rather than to a specific action.

It could be useful, where possible, to hypothesize them, to have a table with possible specific mechanisms of action of vitamin D, regardless of a deficiency.

Author Response

Comments to the Author

The manuscript is well organized and easy to read, covers many aspects, and even the figures and tables are well organized. As pointed out by the authors, there are no studies that clarify whether vitamin D supplementation is necessary and/or beneficial for various diseases.:

Authors: We thank Reviewers 1 for the positive comments, the interest shown in our manuscript and the possibility of reconsidering its publication after revision. We hope that the changes made in the manuscript according to his/her suggestions will have addressed all the points and that the manuscript can be reconsidered for publication.

Comments to the Author

Probably, it should be emphasized that very often in the general population, especially in the winter seasons, a deficiency of vitamin D is noted; therefore, being some actions being well established (immune system in primis, but also inflammatory state), the correlated positive actions on the pathologies may be due only to the restoration of "normal" values rather than to a specific action. It could be useful, where possible, to hypothesize them, to have a table with possible specific mechanisms of action of vitamin D, regardless of a deficiency.

Authors: Accordingly, we added a sentence to emphasize the VitD deficit in the general population and the implication of the restoration of regular values.

Reviewer 2 Report

General comment:

This manuscript, entitled “Role of vitamin D in liver disease and complications of advanced chronic liver disease,” authored by Ravaioli et al., reviewed the pathophysiological mechanisms underlying vitamin D deficiency in patients with chronic liver disease to address the different liver etiologies and the condition of advanced chronic liver disease (cirrhosis) with related complications. A descriptive survey helps target catabolic proteins for progressive chronic liver disease. This survey will provide essential links between vitamin D and advanced chronic liver disease. Still, it will also inspire therapeutic approaches for liver dysfunction. In my opinion, this is a valuable work and is suitable for publication in the International Journal of Molecular Sciences after the authors have addressed the following comments and questions:

Specific comments:

1)     Does liver damage cause an overall decrease in several P450 levels and cause a general decrease in several essential or vital components like vitamin D?

2)     Human CYP24A1 has 24 and 23 hydroxylations; how is the ratio of those products (intermediate or end product) involved in the cells' metabolic regulation of vitamin D level?

3)     Suggestion - some recent work. Cite these in 3.2. Metabolism of vitamin D

a) Specificity of the Redox Complex between Cytochrome P450 24A1 and Adrenodoxin Relies on Carbon-25 Hydroxylation of Vitamin-D Substrate. Drug Metab Dispos. 2019 Sep;47(9):974-982. doi: 10.1124/dmd.119.087759.

b) Evidence of Allosteric Coupling between Substrate Binding and Adx Recognition in the Vitamin D Carbon-24 Hydroxylase CYP24A1. Biochemistry. 2020 Apr 21;59(15):1537-1548. doi: 10.1021/acs.biochem.0c00107

4)     How lowering vitamin D levels in any case of the pathological condition can be described?

5)     How can elevated vitamin D levels in any pathological condition be described? Eg - Idiopathic infantile hypercalcemia.

Author Response

Comments to the Author

This manuscript, entitled "Role of vitamin D in liver disease and complications of advanced chronic liver disease," authored by Ravaioli et al., reviewed the pathophysiological mechanisms underlying vitamin D deficiency in patients with chronic liver disease to address the different liver etiologies and the condition of advanced chronic liver disease (cirrhosis) with related complications. A descriptive survey helps target catabolic proteins for progressive chronic liver disease. This survey will provide essential links between vitamin D and advanced chronic liver disease. Still, it will also inspire therapeutic approaches for liver dysfunction. In my opinion, this is a valuable work and is suitable for publication in the International Journal of Molecular Sciences after the authors have addressed the following comments and questions:

 Authors: We thank Reviewer 2 for our manuscript's positive comments and interest and find out the paper suitable for publication.

Comments to the Author

Specific comments:

1)     Does liver damage cause an overall decrease in several P450 levels and cause a general decrease in several essential or vital components like vitamin D?

2)     Human CYP24A1 has 24 and 23 hydroxylations; how is the ratio of those products (intermediate or end product) involved in the cells' metabolic regulation of vitamin D level?

3)     Suggestion - some recent work. Cite these in 3.2. Metabolism of vitamin D –

  1. a) Specificity of the Redox Complex between Cytochrome P450 24A1 and Adrenodoxin Relies on Carbon-25 Hydroxylation of Vitamin-D Substrate. Drug Metab Dispos. 2019 Sep;47(9):974-982. doi: 10.1124/dmd.119.087759.
  2. b) Evidence of Allosteric Coupling between Substrate Binding and Adx Recognition in the Vitamin D Carbon-24 Hydroxylase CYP24A1. Biochemistry. 2020 Apr 21;59(15):1537-1548. doi: 10.1021/acs.biochem.0c00107

Authors: In accordance with the suggestions, we cited the suggested references and added the specifics of the role of liver damage on cytochrome in the manuscript.

Comments to the Author

4)     How lowering vitamin D levels in any case of pathological condition can be described?

5)     How can elevated vitamin D levels in pathological conditions be described? Eg - Idiopathic infantile hypercalcemia.

Authors: On the definition of reduced vitamin D, see the reading of sub-chapter 3.5, "Vitamin D deficiency", where it is well represented. Regarding the pathological conditions of elevated Vitamin D levers, we do not think it would be entirely relevant to our paper and would lead to increased disarray in the flow of reading.
